# Representation from India in multinational, interventional, phase 2 or 3 trials registered in Clinical Trials Registry-India: A cross-sectional study

**Jaishree Mendiratta[1], Ravi N. Vaswani[2], Gayatri Saberwal[1]** *

**1** Institute of Bioinformatics and Applied Biotechnology, Biotech Park, Electronics City Phase 1, Bengaluru, Karnataka, India, **2** Yenepoya Deemed to be University, University Road Deralakatte, Mangaluru, Karnataka, India

* gayatri@ibab.ac.in, gayatri.saberwal@gmail.com

## Abstract

In multinational trials that have run in India, we wished to determine whether there was too much (60% or higher) recruitment from India. We downloaded all trial records from Clinical Trials Registry-India, CTRI, and stored them in a local SQLite database. We queried records registered in a recent 8-year period, ie 2013–2020 and evaluated the fraction of local participants in interventional Phase 2 or Phase 3 studies. 62 trials were completed, with completion dates available. Five trials (8%) had 60% or more *planned* recruitment from India. Four of the five (7% of 62) had a foreign sponsor, and therefore there was an unfair burden-benefit ratio on the Indian population. Seven trials (11%), of which six (10% of 62) had foreign sponsors, had 60% or more (of the total) *actual* recruitment from India, and for two trials (both with foreign sponsors), the data were meaningless. There were 362 studies that were listed as not completed, although, given their start date and estimated duration, some of them ought to have been. Twenty five cases (7% of 362) had 60% or more *planned* recruitment from India. Of these, 18 (5% of 362) had foreign sponsors and were potentially problematic. Even allowing for some delays in completion, 128 (35% of 362) studies ought to have been completed by the time of our study. As such, we identified several problematic trials for which the *planned* recruitment from India in multinational studies was 60% or more. We also identified trials in which the *actual* recruitment was significantly higher than the *planned* recruitment. Further, the records of several studies that were probably completed were not updated in CTRI in a timely manner. The Indian drug regulator needs to be particularly alert to the *planned*, or *actual*, over-recruitment of participants from India. Further, CTRI, alone or in collaboration with the regulator, needs to ensure that multinational trial records for the enrollment fields in particular are updated, in a timely manner.

**Data Availability Statement:** S2 File and S3 File are available at https://osf.io/bsncw and at https://osf.io/kr9f8, respectively. The rest of the data used

for this study are available in the main text or the additional files.

**Funding:** This work was supported by internal funding of the Institute of Bioinformatics and Applied Biotechnology, from the Department of Electronics, IT, BT and S&T of the Government of Karnataka. The funder had no role in study design, data collection and analysis, decision to publish, or preparation of the manuscript.

**Competing interests:** The authors have declared that no competing interests exist.

**Abbreviations:** CDSCO, Central Drugs Standard Control Organisation; CRO, Contract Research Organization; CTRI, Clinical Trials Registry - India; US, United States.

## Introduction

Until 2005, multinational clinical trials in India had to be one phase behind the phase running elsewhere. Phase parity was established in 2005 [1], leading to an explosion in the number of studies running in the country [2]. The rising number of multinational trials in India was part of the broader trend of the globalization of studies, which were increasingly moving out of 'traditional' trial nations such as the United States (US) and Western Europe, to 'non-traditional' ones such as Eastern Europe, Russia and Asia [3]. Various factors, such as (i) reduced costs in the new locations; (ii) tougher regulations in the US; (iii) the availability of 'treatment naive' patients in the new locations; (iv) the greater speed of enrollment in the new locations; and (v) a step to obtain marketing approval in these countries in future, drove the process [4]. However the move was also criticized on multiple counts such as (i) studies being conducted for diseases that were not the most pressing in the new locations; (ii) the possibility that poverty could lead to the undue inducement of participating in trials; (iii) the questionable practice of using Western informed consent frameworks in other societies; (iv) the ethics of conducting studies in populations that would not always be able to afford the new drugs [5]; (v) lack of clarity on the experience of the trialists in the new locations [4]; and (vi) the uncertainty about the generalizability of results obtained in the new populations [4].

Unfortunately, the rise in the number of trials in India was followed by controversies such as participants not providing informed consent, or not being compensated for serious adverse events [2]. Also, there have been cases in the past where about half the recruits in international studies were *planned* to be recruited from India, but in fact more than 80% were [6]. All of this has led to apprehensions that people in developing countries have sometimes been exploited in trials sponsored by organizations in the developed world [6–9]. However this question has never been systematically investigated. Therefore, for multinational trials that have run in India, we set out to determine whether there was too much (60% or higher) recruitment from India.

## Methods

In India, trials are registered with Clinical Trials Registry-India (CTRI) [10]. Sample studies registered with CTRI, and their URLs, are available in S1 File. On 28 January 2022, we downloaded all the trials in CTRI using a script that we developed in R programming language (available at https://osf.io/bsncw as S2 File). The script web scraped and cleaned the data, and stored them in an SQLite database (available at https://osf.io/kr9f8 as S3 File). S4 File contains the schema of the database.

Of the 39,821 downloaded trials, we examined the 26,889 that were registered with CTRI between 1 Jan 2013 to 31 Dec 2020, inclusive (S5 File). It is known that older CTRI records had more errors [11] and that is why we focused on a more recent time period.

After the processing of these trials to obtain the sub-set of interest, we identified our main outcome variable, that is participation from India. We did this by examining either the *planned* fraction of recruitment from India, that is 'Sample Size from India'/'Total Sample Size', or the *actual* fraction of recruitment from India, that is 'Final Enrollment numbers achieved (India)'/ 'Final Enrollment numbers achieved (Total)'.

Since we wished to determine whether there was too much recruitment from India, we defined 'high recruitment' as 60% or higher recruitment from India. Here we briefly mention our reason for choosing 60% as a cut off. Trials come with risks for the participants. They also come with potential (a) immediate benefit to the participants, or (b) subsequent benefit to those of the same ethnicity in case the intervention is subsequently approved and is affordable and accessible to similar populations. In a multinational trial, the fraction of participants from India should never be more than 50%, and proportionately lower based on how many

countries participate in the trial. However, given that India's population is almost 1.4 billion people, which implies the ready availability of patients, and the fact that it ought to be easier and quicker to conduct a trial in fewer countries, we believe that recruiting up to 60% of the participants from India is ethical.

## Results

The processing of these 26,889 trials is shown in Fig 1. We used the field *Type of trial* to identify the subset of 19,361 interventional studies and *Phase* to identify the subgroup of 6437 phase 2 and phase 3 trials (S5 File). Next, we used the following pair of fields to identify the multinational studies: *Recruitment Status of Trial (Global)* and *Recruitment Status of Trial (India)*. So, for instance, if either field was 'not applicable', then the study was rejected. However, given the types of errors that are known to be present in the CTRI records [11], we also used the following two pairs of fields to identify the studies that were unambiguously multinational: (a) *Date of First Enrollment (India)* and *Date of First Enrollment (Global)* and (b) *Total Sample Size* and *Sample Size from India*. For the trial to be considered multinational, in (a) both fields needed to have values, and the total sample size had to exceed the sample size from India. Thus, a trial was classified as Indian or multinational only if the data in all three pairs of fields were consistent with the relevant classification.

Based on these criteria, we obtained 5580 Indian studies, 473 multinational ones and 384 other trials (S6 File). Of the 384 other cases, 22 were entirely foreign trials; 77 were terminated either in India or elsewhere; 1 was suspended in India; 256 appeared to be Indian trials based on data in some fields, but had ambiguous information in other fields; 18 appeared to be multinational trials based on data in some fields, but had ambiguous information in other fields; and 10 trials had other kinds of discrepancies. We provide three examples from these 384 cases in Table 1.

We took forward the 473 multinational studies.

We first wished to identify completed trials, and chose those for which data were available both for '*Date of Study Completion (India)*' and '*Date of Study Completion (Global)*'. We obtained 62 studies, which we termed the Completed Set (S7 File). Next, we selected cases where both '*Date of Study Completion (India)*' and '*Date of Study Completion (Global)*' listed 'Applicable only for Completed/Terminated trials'. That is, these studies were not completed. We called these 362 trials the Not Completed Set (S8 File).

We then took the following steps to estimate the date by which the Not Completed Set studies ought to have been completed and the study details updated in the CTRI records, while allowing some margin for delays.

i.  Of the '*Date of First Enrollment (India)*' and '*Date of First Enrollment (Global)*', we considered the later date as the overall '*Date of First Enrollment*'.

ii. We then examined the '*Estimated duration*' of the trial. This value is expressed in years, months and days. Where 'days' had a non-zero value, we substituted it with one month. Also, we converted the years to months, so that the final 'Estimated duration' was expressed in months.

iii. In order to allow the sponsor a margin of time to update the registry record in case the trial was delayed, (a) we doubled the number of months (and termed it '2x months') for studies whose estimated duration was up to two years; and (b) increased the number of months 1.5 times (termed '1.5x months') for studies whose estimated duration was 2–5 years. We did not provide such a margin for trials whose estimated duration was longer than five years.

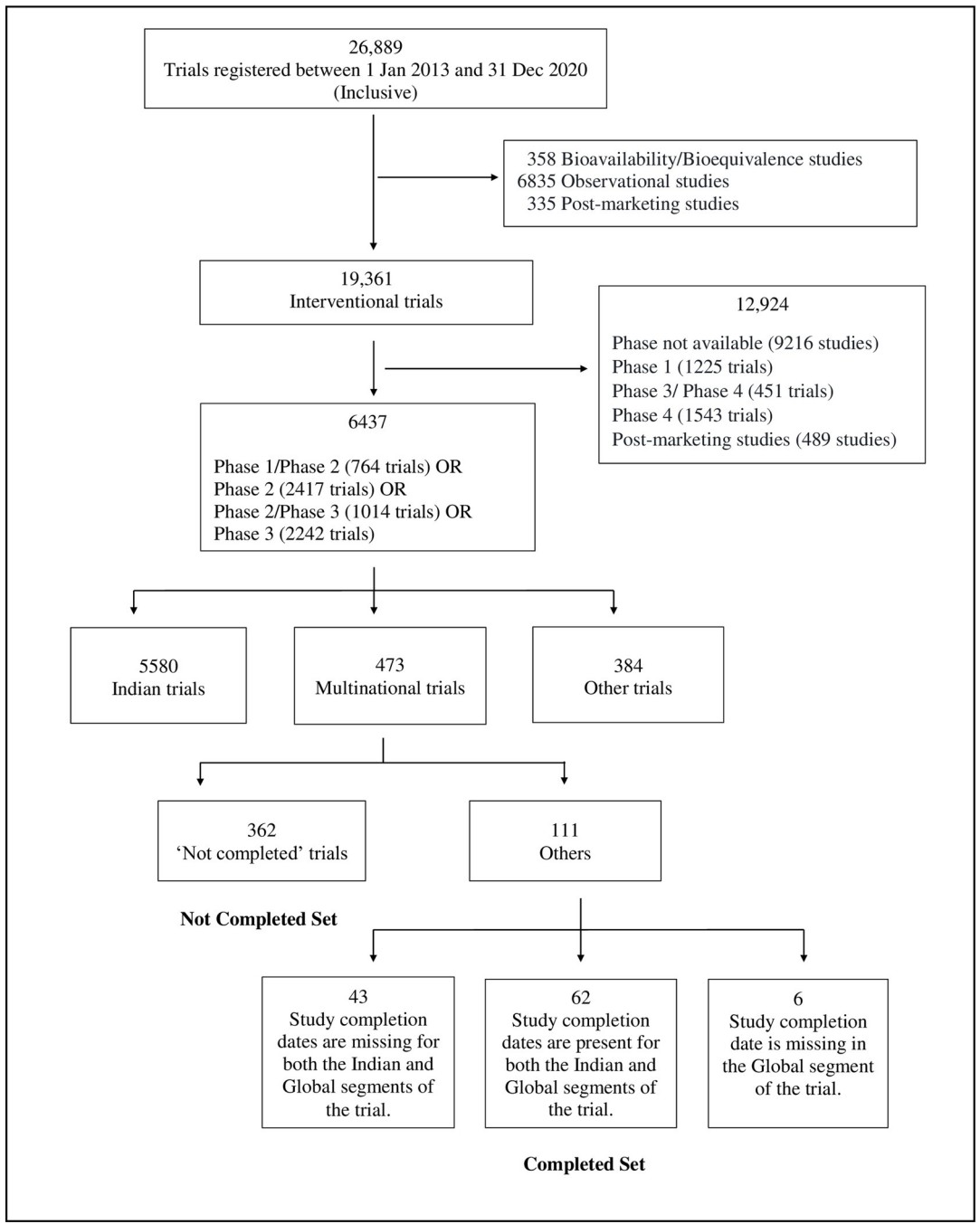

**Fig 1. The processing of the 26,889 trials registered with CTRI between 1 Jan 2013 and 31 Dec 2020.**

iv. We added the number of months obtained from step (iii) to the overall '*Date of First Enrollment*' from step (i), to yield the estimated '*Date of Study Completion*'.

v. In order to give the sponsor time to update the CTRI record, we added a further 6 month margin to the estimated '*Date of Study Completion*' for each trial, to obtain the date of study filing with the registry.

**Table 1. Three samples from the 384 trials that had ambiguous data, and that were excluded from this study.**

| | CTRI Number | Total Sample Size | Sample Size from India | Recruitment Status_Global | Recruitment Status_India | Date of first enrollment_Global | Date of first enrollment_India |
|---|---|---|---|---|---|---|---|
| 1 | CTRI/2013/02/003388 | 700 | 700 | Not Applicable | Not Applicable | 15/02/2016 | Date Missing |
| 2 | CTRI/2019/06/019635 | 588 | 588 | Completed | Completed | 01/08/2019 | 20/06/2019 |
| 3 | CTRI/2020/06/026230 | 40 | 20 | Not Applicable | Completed | Date Missing | 07/08/2015 |

*Example 1*: If there is recruitment from India, as 'Sample size from India' states, then 'Recruitment Status_India' cannot be 'Not applicable'. And if it is not a multinational trial (since the 'Total Sample Size' is the same as 'Sample size from India', then there cannot be any data in the field 'Date of first enrollment_Global'.

*Example 2*: If 'Total Sample Size' is the same as 'Sample size from India', it implies that this is an Indian trial. If so, then 'Recruitment Status_Global' cannot be 'Completed', and 'Date of first enrollment_Global' should not have a date.

*Example 3*: If 'Total Sample Size' is greater than 'Sample size from India', it implies that this is a Multinational trial. If so, then 'Recruitment Status_Global' cannot be 'Not applicable' and 'Date of first enrollment_Global' should not be 'Date missing'.

This date of study filing gives the sponsor reasonable time to complete the trial and update the CTRI record. We went on to identify how many trials in the Not Completed Set had not updated their records by this date of study filing (S9 File).

We posed three questions of the Completed Set and two questions of Not Completed Set, as described below.

## 62 trials of completed set: Planned recruitment from India

We first considered the 62 studies of the Completed Set, that is trials that were completed and with data available for both '*Date of Study Completion (India)*' and '*Date of Study Completion (Global)*'. We examined the fraction of recruitment that was *planned* to be from India, and identified five cases (8% of 62) with 60% or more *planned* recruitment.

One trial's primary sponsor was a domestic pharma company, and the trial (CTRI/2017/11/010555) was concerned with cardiovascular risk. Another study (CTRI/2017/09/009699) had a foreign sponsor, that directly worked with a well-known eye hospital in India. The trial was concerned with uveitis. Finally, three studies (CTRI/2020/02/023099, CTRI/2020/06/026192 and CTRI/2020/08/027248) were sponsored by foreign entities that worked through Contract Research Organizations (CROs). These trials were concerned with schizophrenia, COVID-19 infection and hemorrhoids respectively. As such, four of the five (7% of 62) trials had a foreign sponsor. Further details are available in S7 File.

## 62 trials of the completed set: Actual recruitment from India

Next, for the 62 studies of the Completed Set, we examined the *actual* fraction of recruitment from India. We identified seven cases (11% of 62) with 60% or more (of the total) *actual* recruitment and two cases where the data could not be used. Among the seven cases, four (CTRI/2017/11/010555, CTRI/2020/02/023099, CTRI/2017/09/009699, CTRI/2020/06/026192) have been described above. Details of the three remaining cases were as follows: CTRI/2014/09/004965 was for breast cancer and CTRI/2018/05/013650 for macular degeneration (both sponsored by Samsung Bioepis Co Ltd, South Korea); and CTRI/2018/02/011794 was for HIV (National Institute of Allergy and Infectious Diseases, National Institutes of Health, US). Of the seven, six cases (10% of 62) had foreign sponsors. In the two cases of problematic data, CTRI/2017/12/010935 (sponsored by Amgen Inc., US) and CTRI/2018/01/

011307 (AstraZeneca AB, Sweden), the enrollment from India exceeded total enrollment. Further details are available in S7 File.

For both the *planned* and *actual* recruitment, we carried out a sensitivity analysis of 60% +/- 5% and 60% +/- 10% (S10 File). There were very small changes in the number of 'problematic' trials with the altered cut offs.

## 62 trials of the completed set: Ratio of actual to planned percent recruitment from India

Finally, for the Completed Set, we examined the ratio of the *actual* percent recruitment from India to the *planned* percent. Cases with a ratio greater than 1.0, and cases with incorrect data, are described below, and are available in Table 2 and S7 File.

These cases fall into four groups, as follows:

a. In six cases, *actual* enrollment was fairly close to *planned* recruitment, being up to 15% higher.

b. In two cases, *actual* enrollment was 16–60% higher than the *planned* recruitment.

c. The five cases in which the ratio of *actual* to the *planned* percent exceeded 1.6 are as follows. Three cases were listed in the previous section. To recapitulate, CTRI/2014/09/004965 was for breast cancer and CTRI/2018/05/013650 for macular degeneration (both sponsored by Samsung Bioepis Co Ltd, South Korea); and CTRI/2018/02/011794 was for HIV (National Institute of Allergy and Infectious Diseases, National Institutes of Health, US). The details of the other two studies are as follows: CTRI/2015/07/005984 for myocardial injury (Hamilton Health Sciences Corporation through its Population Health Research Institute, Canada and CTRI/2020/06/026192 for COVID-19 infection (Vicore Pharma AB; a Swedish pharma company). Notably, in three of these five cases a *planned* fraction of recruitment from India of between 12 and 18% changed to 100%.

d. Of the four trials whose data could not be used, two were described in the previous section: CTRI/2017/12/010935 (sponsored by Amgen Inc, US) and CTRI/2018/01/011307 (AstraZeneca AB, Sweden). In these cases, the enrollment from India exceeded total enrollment. In the other two cases, CTRI/2018/02/011983 (Eli Lilly and Company India Pvt Ltd, India); and CTRI/2018/02/012091 (mAbxience, Spain), the total enrollment was listed as 0.

## 362 trials of the not completed set: Planned recruitment from India

We went on to consider the 362 studies in the Not Completed Set, that is cases where both the '*Date of Study Completion (India)*' and the '*Date of Study Completion (Global)*' listed 'Applicable only for Completed/Terminated trials'. That is, these studies appeared not to be completed

**Table 2. Trials for which (i) the ratio of *actual* to *planned* percent recruitment from India was above 1, or (ii) incorrect data prevented the calculation of this ratio.**

|  | The ratio of *actual* to *planned* percent recruitment from India | Number of trials (%) |
|---|---|---|
| 1. | *Actual* enrollment was 1.01–1.15 x *planned* enrollment | 6 (35.3%) |
| 2. | *Actual* enrollment was 1.16–1.60 x *planned* enrollment | 2 (11.8%) |
| 3. | *Actual* enrollment was 1.61 and above x *planned* enrollment | 5 (29.4%) |
| 4. | Incorrect data | 4 (23.5%) |
|  | **TOTAL** | **17 (100%)** |

(S8 File). We determined the fraction of recruitment that was *planned* to be from India, and identified 25 cases (7% of 362) for which this fraction was 60% or more (Table 3 and S8 File).

Of these 25 studies, three were sponsored by Indian companies. Another one was by a multinational firm that acquired a local company that was involved with this trial. One study was sponsored by a public sector research institute set up by the Government of India. Two studies concerned conditions (pediatric tuberculous meningitis and post kala azar dermal leishmaniasis) that are more prevalent in India than in the West. Eight trials were sponsored by non-Western companies located in Brazil, Malaysia or Russia. The remaining 10 studies were sponsored by companies based in Australia, Austria, Italy, Switzerland, or the US. The latter 18 studies related to conditions that are not preferentially prevalent in India. As such 18 trials (5% of 362) had foreign sponsors that were potentially problematic.

### 362 trials of the not completed set: Data in CTRI not updated in time

Finally, we examined the number of 'not completed' studies for which '*Date of Study Completion (India)*' and '*Date of Study Completion (Global)*' data ought to have been updated in CTRI, but had not been. Since we downloaded trial data on 28 January 2022, we determined the number of trials whose date of study filing, after we had given several margins for the completion of the study and six months for updating the record in CTRI, was by 27 January 2022, inclusive. The records of 128 (35% of 362) studies ought to have been updated by this date (S9 File).

## Discussion

This work is concerned with the over-recruitment of participants from India in international trials. Undoubtedly trials conducted in India would have undergone review by Ethics Committees. However it is known that the processing of a trial proposal by an ethics committee may be perfunctory [12]. All trials in India have also received regulatory permission, if relevant. Here, too, it is known that permission may have been granted in a casual manner [13]. And, as mentioned earlier, both Indian and foreign researchers have commented on the ethical transgressions that have taken place in some trials run in India. For the 62 completed trials most of the trials with 60% or more *planned* or *actual* recruitment had foreign sponsors. Likewise, for the 362 trials that were listed as not completed, most of the cases with 60% or more *planned* recruitment had a foreign sponsor. Given the history of exploitation of trial participants in India, and given that all trials carry an element of risk, we believe that there is an unfair burden-benefit ratio on the Indian population if foreign sponsors, that are conducting trials in multiple countries, recruit too many patients from this country.

**Table 3. Some details of the 25 'not completed' trials where the *planned* recruitment from India was 60% or more of the total recruitment.**

| No. | Categories of sponsors, or conditions | Number of trials (%) |
|---|---|---|
| 1. | The sponsor was an Indian company | 3 (12%) |
| 2. | The sponsor was a multinational company that acquired a local company that was involved with this trial | 1 (4%) |
| 3. | The sponsor was an Indian public sector research institute | 1 (4%) |
| 4. | The conditions under investigation are perhaps more prevalent in India | 2 (8%) |
| 5. | The sponsors were from Brazil, Malaysia or Russia | 8 (32%) |
| 6. | The sponsors were organizations based in the West, for conditions found around the world | 10 (40%) |
| | **TOTAL** | **25 (100%)** |

Before coming to an analysis of particular studies, we wish to comment on the low number of trials included in this study. It is known that trialists may register an interventional study as an observational one [11]. It is also known that records may have internal discrepancies of phase etc. As such, even though we used various filters in order to identify a well-defined subset of trials, we may have inadvertently either included a trial that we did not wish to, or excluded a valid trial. Any such error is a function of errors in the trial records. Having used the filters that we did, the number of trials came down significantly, leading to a dataset of just a few 100 trials.

We sought to determine whether, in multinational studies that had run in India, the fraction of Indian participants was *actually* high, or had been *planned* to be high. In Methods, we defined 'High' as 60% or higher recruitment from India, and explained the reasons thereof. In a multinational trial, where there is at least one country other than India, there ought to be a fair distribution of both the risks and potential benefits to the participants in the participating countries. As such, the fraction of participants from India should never be more than 50%. However, in the interest of each trial being conducted in a realistic time-frame, and without the extra costs of setting up sites in too many countries, we believe that recruiting 60% of the participants from India is ethical. We also found that doing a sensitivity analysis around 60% revealed a predictability that lends credence to this premise.

In this work, we have considered two sets of trials, ie (i) completed studies for which the data of final enrollment were available (Completed Set), and (ii) studies that are classified as not completed (Not Completed Set).

With regard to the Completed Set, we first identified five cases with 60% or more *planned* recruitment from India. Since one study (for cardiovascular risk) was sponsored by a domestic pharma company, without the participation of any foreign entity either as secondary sponsor or as 'other source of monetary or material support', the high *planned* recruitment from India was not a concern. The remaining four were concerned with uveitis, schizophrenia, COVID-19 infection and hemorrhoids. Given that these four conditions are well distributed around the world, we believe that planning to recruit such a large fraction of trial participants from India was not justified.

Next, for the Completed Set, we examined the ratio of *actual* to *planned* percent recruitment from India. Viewed from the view point of potential exploitation of participants from India, we believe that it was worth noting whether the *actual* recruitment was more than the *planned*, and if so, to what extent. In examining the extent of over-recruitment, we needed to use some arbitrary cut offs for what increase could be considered acceptable. We realize that if more patients are identified at some trial sites, who are willing to be part of the trial, then for practical reasons of seeing the trial progress in a timely manner, it is convenient to allow increased enrollment from those sites. We believed it acceptable if the *actual* was up to and including 15% of the *planned* recruitment. Only six (35% of the 17 potentially or actually problematic) trials were in this category, which is quite low. We considered it questionable if the *actual* was 16–60% higher than the *planned* recruitment, and we found it unacceptable if the *actual* was over 60% higher than the *planned*. These two categories–between 16–60% and over 60%–are arbitrary, and other researchers may prefer other cut offs. However, as noted, in three cases the recruitment from India rose from under 20% to 100%. Therefore, even changing the cut off would not alter the fact that certain 'multinational' trials became 'Indian' trials during the course of the trial. The Indian regulator, that is the Central Drugs Standard Control Organisation (CDSCO), needs to take cognizance of such increases in fraction of enrollment from India, and issue guidelines or rules on what is permissible.

The cases where we could not use the data were also problematic. In earlier publications [11, 14], we documented some of the fields for which CTRI records have missing or misleading

data. To be noted, similar issues of missing data have been reported for the registry of the US, ClinicalTrials.gov, as well [15, 16]. The implications of such erroneous information vary with the field. For multinational trials, missing or out-of-date information in the fields '*Date of Study Completion (India)*', '*Date of Study Completion (Global)*', *Total Sample Size* and *Sample Size from India*, prevents audits of the fraction of trial participants recruited from India. Since over-recruitment from India is an issue of concern, it is important that CTRI or CDSCO, finds a way to ensure that these fields are updated in a timely manner.

With regard to the Not Completed Set trials, we first identified the 25 studies for which the *planned* recruitment from India was 60% or more of the total enrollment. It seemed perfectly justified that a large fraction of participants was recruited from India, in situations where (i) the sponsors were local companies (or a multinational company that had taken over a local company, and the local unit was involved in the trial), (ii) the sponsor was a local public sector research institution or (iii) the conditions being investigated were particularly prevalent in India. However, it is difficult to justify trials that were sponsored by those based in Brazil, Malaysia or Russia, since these are populous nations, and it ought to be possible to find trial participants domestically. Likewise, organizations based in Western nations ought to be able to find recruitees in many countries if the medical condition is found around the world.

Further, in the Not Completed Set studies, neither the global nor the Indian '*Date of Study Completion*', had been updated in 362 (77% of the 473) CTRI records. Clearly, some of these were ongoing studies. However it was clear that others must have been completed, but that the CTRI records had not been updated to reflect this. This is despite that fact that the New Drugs and Clinical Trials Rule requires the status of a trial, such as whether it is ongoing or completed, to be reported to the Regulator every six months [17]. Although there do not appear to be any guidelines for updating this field at CTRI, in the US there is a requirement that the trial record at ClinicalTrials.gov be updated within 30 calendar days of the completion of a trial [18, 19]. In the interest of transparency, it would be advisable for CTRI to have a similar rule.

Here, too, we had to define arbitray margins, as outlined earlier, in order to identify trials that were most likely to have been completed. Some of the trials that we identified as 'completed' are likely to be false positives. There are also likely to be some false negatives. Although other researchers may have changed some of these margins, they too will have some false positives or negatives. Given the absence of 'Date of Study Completion' data, it is impossible to correctly identify which trials have actually been completed. Nevertheless, we believe that it is important to provide some estimate of completed trials for which completion dates have not been listed, to establish the phenomenon, and its relevance to the issue of recruitment from India.

Even after providing various types of margins for each record to be updated, we found that 128 (27%) records had not been updated. Due to the large fraction of outdated records, there was no way to determine–for these trials–whether the phenomenon of excess recruitment from India was common or not. The large fraction of outdated records is cause for concern, and CTRI needs to find ways to ensure that they are updated. Although it is known that registry staff check the records when they are first submitted [20], it is not known whether they check updates to records. Nevertheless, there must be best practices followed by the highly rated public registries such as ClinicalTrials.gov; Clinical Research Information Service, Republic of Korea; Australian New Zealand Clinical Trials Registry; Iranian Registry of Clinical Trials; and the German Clinical Trials Register [21], which CTRI could learn from to ensure updated study records.

Finally, we wish to point out that not only is over-representation from India a potential problem, but under-representation is also a problem, and we have highlighted the latter issue recently [22]. However the reasons that over- and under-representation are problematic are different. In the former situation, it is the worry of exploitation of Indian participants, and in the

current work, we have focussed on this aspect. In the latter situation, it is that without adequate ethnic diversity, the trial results may not be generalizable to a larger population. Further, for many patients, drugs for rare disease can be completely unaffordable, and participation in a trial provides a way to potentially benefit from a novel therapy. So, although under-representation in multinationals is also a potential problem, we did not focus on that angle in this study.

## Limitations

The study had a few limitations, as follows: (i) Although we selected for 'interventional', 'phase 2' and 'phase 3' trials, we may not have captured all of such studies due to the mislabelling of particular records [11]. As such, we do not know whether the trends that we have noted applied to the entire body of relevant trials. Also, as a result, the study was based on a small number of trials. (ii) We have allowed up to 60% recruitment to take place from India without passing an adverse comment. Some researchers may prefer a lower limit. If so, we may have missed a few cases of over-recruitment from India. (iii) Since most studies had not updated their date of completion, our dataset was, again, truncated. (iv) The multiple 'margins' that we allowed for each study record to be updated were arbitrary. Therefore, we may have incorrectly labelled some trials as 'completed' and therefore whose estimated Global and Indian '*Dates of Study Completion*' had been reached although they were not, or missed some trials whose reporting dates had indeed been reached.

## Conclusion

In summary, for trials registered in a recent eight year period, we have identified several problematic trials for which the *planned* or *actual* recruitment from India in Phase 2 or Phase 3 multinational studies was 60% or more. We also identified trials in which the *actual* recruitment was significantly higher than the *planned* recruitment. Further, the records of several studies that were probably completed were not updated in CTRI in a timely manner.

It is not our intention to question the practice of running multinational trials in India. Nor are we inferring malafide intentions of the CTRI staff, or of CDSCO. We merely wish to point out that CDSCO should be particularly alert to the *planned*, or *actual*, over-recruitment of participants from India. Further, CTRI, alone or in collaboration with CDSCO, needs to ensure that multinational trial records are updated for the enrollment fields in particular, in a timely manner.

## Supporting information

**S1 File. CTRI numbers of sample trials registered with the Indian registry CTRI, and the URLs at which they are available.**
(DOC)

**S2 File. The R script used to download data from CTRI, process them and store them in an SQLite database.**
(DOCX)

**S3 File. Details of 39,821 records from CTRI, stored in an SQLite database.**
(DOCX)

**S4 File. The schema of the SQLite database.**
(DOCX)

**S5 File. The steps taken to identify all the trials registered with CTRI between 1 January 2013 and 31 December 2020, that were interventional, and phase 2 or phase 3.**
(XLSX)

**S6 File. The steps taken to identify trials that were unambiguously Indian or multinational trials, and those that were ambiguous, and the lists of these three groups of trials.**
(XLS)

**S7 File. Identification of the 62 multinational trials, completed in India and elsewhere, and details of the planned and actual recruitment of these 62 trials.**
(XLS)

**S8 File. The 362 'not completed' trials, and 25 of the 362 trials that had 60% or more planned recruitment from India.**
(XLS)

**S9 File. The 'not completed' 362 cases where both 'Date of Study Completion (India)' and 'Date of Study Completion (Global)' listed 'Applicable only for Completed/Terminated trials', and their break up based on 'Estimated duration of trial' into short, intermediate and long subsets.**
(XLS)

**S10 File. A sensitivity analysis for the 60% cut off for over-recruitment from India.**
(XLS)

## Author Contributions

**Conceptualization:** Gayatri Saberwal.

**Data curation:** Jaishree Mendiratta.

**Formal analysis:** Jaishree Mendiratta, Ravi N. Vaswani, Gayatri Saberwal.

**Funding acquisition:** Gayatri Saberwal.

**Investigation:** Jaishree Mendiratta, Gayatri Saberwal.

**Methodology:** Ravi N. Vaswani, Gayatri Saberwal.

**Project administration:** Gayatri Saberwal.

**Resources:** Gayatri Saberwal.

**Software:** Jaishree Mendiratta.

**Supervision:** Gayatri Saberwal.

**Validation:** Gayatri Saberwal.

**Visualization:** Gayatri Saberwal.

**Writing – original draft:** Gayatri Saberwal.

**Writing – review & editing:** Jaishree Mendiratta, Ravi N. Vaswani, Gayatri Saberwal.

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
