## [Decision Letter · Decision Letter 0]

26 Sep 2022

PONE-D-22-19988Participants from India are over-represented in some multinational trialsPLOS ONE

Dear Dr. Saberwal,

Thank you for submitting your manuscript to PLOS ONE. After careful consideration, we feel that it has merit but does not fully meet PLOS ONE’s publication criteria as it currently stands. Therefore, we invite you to submit a revised version of the manuscript that addresses the points raised during the review process.

The manuscript has been evaluated by two reviewers, and their comments are available below.

The reviewers have raised a number of major concerns including that they feel the manuscript should outline clear justification for the 60 % criterion along with sensitivity analyses. 

Could you please carefully revise the manuscript to address all comments raised?

We look forward to receiving your revised manuscript.

Kind regards,

Alice Coles-Aldridge

Editorial Office

PLOS ONE

Journal Requirements:

Reviewers' comments:

Reviewer's Responses to Questions

**Comments to the Author**

1. Is the manuscript technically sound, and do the data support the conclusions?

Reviewer #1: Yes

Reviewer #2: Yes

2. Has the statistical analysis been performed appropriately and rigorously? 

Reviewer #1: Yes

Reviewer #2: Yes

3. Have the authors made all data underlying the findings in their manuscript fully available?

Reviewer #1: Yes

Reviewer #2: No

4. Is the manuscript presented in an intelligible fashion and written in standard English?

Reviewer #1: Yes

Reviewer #2: Yes

5. Review Comments to the Author

Reviewer #1: The authors raise a valid concern over multinational trials being selectively done in India. This is an area that needs exploration.

However, one concern that I have is that per se, having higher patients from India is not a bad thing. Authors need to make this clearer. What is their main concern with India being overrepresented in the trials and why? As long as this is done a priori with all necessary permissions from Indian authorities. Similarly, increasing participation from India later in the trial also should be OK, if the processes are followed.

Today Indian companies want to make drug for the world and generally licensing in the home state is a requirement. So, what is wrong if they make medicines for conditions which are not “preferentially prevalent in India”.

There are definitely problems with the CTRI website and its updation. Are these problems only for multinational trials OR are also seen in Indian trials. This is important before we infer malafide intentions.

Comments:

Title: needs rephrasing as “some” should not be cause of concern. Better to be neutral – “Representation from India in multinational trials registered in national trial registry”.

Authors do not explain how they defined unfair burden-benefit ratio. Similarly, they need to define their main outcome variable (participation from India) and its measurement in the methods section.

There is poor differentiation between methods/ results and discussion. Needs to be rewritten. Results starts with a statement – we posed three questions which is clearly a methods component. Issue of Sanofi Pasteur and Shanta Biotech – if at all it needs to be in the paper, it should be in discussion.

For example, figure 1 is a part of result and not methods.

Might be better to give details of the trials (table 1) in supplement.

There are many arbitrary definitions and cut-offs which need to be justified. 60% is an example. 60% or more actual recruitment – was this 60% out of total or 60% more than the planned (denominator is the planned number).

I would not put not studying before 2013 as a limitation.

Figure 1. Should be like a typical flow charts and explain exclusions as well. It would be useful if data on all phases of trials are provided before picking phase 2-3 trials.

Reviewer #2: Thank you for the opportunity to review this paper. The issue of selective recruitment and representativeness of trials is of increasing importance. I think it could be of interest to more general readers of PLOS

The paper needs a structured abstract, and the raw numbers in the abstract would benefit from percentages.

The title might need to give a clearer sense of the types of trials included and should probably give a sense of the study design for the interested reader.

I would move the reason for the eight year period to the methods, as I felt that justification for that was only

introduced late in the paper.

What was an ‘ambiguous’ trial (page 5). Some examples might be useful here as there are lots of these and they are not really considered further in the analysis.

The study is largely desccriptive and I think the methods are sensible and appropriate, with the main issues around the exact procedures used to assign trials to various categories.

I do think the reason for the 60% criterion needs justification. As they highlight, it may be too low, but they need to give a sense of the reasons behind that threshold, as it is not clear why that was chosen. There is a good argument for some sensitivity analyses around this figure, since it is so important. To avoid confusing the main message, they might be included as an ‘additional file’ for interested readers to show how different thresholds impact on the results.

Labelling (and naming) studies as ‘problematic’ or ‘ethically questionable’ may be an issue. I assume all these studies have received ethical clearance of some form, and although I do not feel that means that they cannot be reconsidered in terms of ethical issues by other observers, it is quite a leap to apply such labels, without the authors/companies having right of reply. Over-recruitment in particular settings does raise ethical issues, but I do not think it is by definition ‘unethical’ or ‘questionable’ or ‘unacceptable’ (which are quite loaded terms), and I think that is an important distinction. I would use less emotive language in labelling. The ethical issues can be highlighted in the introduction and discussion.

The labelling of studies as Set A and Set B is not very helpful, and it would be much better to have succinct but meaningful labels for these based on the actual criteria used to define them.

6. PLOS authors have the option to publish the peer review history of their article (what does this mean?). If published, this will include your full peer review and any attached files.

Reviewer #1: **Yes: **Anand Krishnan

Reviewer #2: **Yes: **Peter Bower

---

## [Author Response · Author response to Decision Letter 0]

9 Oct 2022

Our responses to the reviewers’ comments:

Reviewer #1: 

1. However, one concern that I have is that per se, having higher patients from India is not a bad thing. Authors need to make this clearer. What is their main concern with India being overrepresented in the trials and why? As long as this is done a priori with all necessary permissions from Indian authorities. Similarly, increasing participation from India later in the trial also should be OK, if the processes are followed.

[We have added the following lines to the Discussion.

“This work is concerned with the over-recruitment of participants from India in international trials. Undoubtedly trials conducted in India have undergone review by Ethics Committees. However it is known that the processing of a trial proposal by an ethics committee may be perfunctory [12]. All trials in India have also received regulatory permission. Here, too, it is known that permission may have been granted in a casual manner [13]. And, as mentioned earlier, both Indian and foreign researchers have commented on the ethical transgressions that have taken place in some trials run in India. For the 62 completed trials most of the trials with 60% or more planned or actual recruitment had foreign sponsors. Likewise, for the 362 trials that were listed as not completed, most of the cases with 60% or more planned recruitment had a foreign sponsor. Given the history of exploitation of trial participants in India, and given that all trials carry an element of risk, we believe that there is an unfair burden-benefit ratio on Indian recruitees if foreign sponsors, that are conducting trials in multiple countries, recruit too many patients from this country.”

We have commented on burden-benefit ratio in our response to Comment 5.]

- - - - - - - - - - - - - - - - - - - - - - - - - - - - - - - - - - - - - - - - - - 

2. Today Indian companies want to make drug for the world and generally licensing in the home state is a requirement. So, what is wrong if they make medicines for conditions which are not “preferentially prevalent in India”.

Our (slightly revised) sentence is as follows: “Eight trials were sponsored by non-Western companies located in Brazil, Malaysia or Russia. The remaining 10 studies were sponsored by companies based in Australia, Austria, Italy, Switzerland, or the US. The latter 18 studies related to conditions that are not preferentially prevalent in India. As such 18 trials (5% of 362) had foreign sponsors that were potentially problematic.” So, we are not judging what Indian companies should or should not do. We are only saying that if a condition is not preferentially present in India, then there is no reason for a foreign sponsor to have very high recruitment from India. Let the foreign sponsors recruit participants from multiple countries, and keep the recruitment from India at less than 60%. (Please also refer to our justification of 60%, that is now added in the beginning of the Discussion.)

To illustrate why we object, we provide (for the reviewer, not in the manuscript) two quotes:

1. A quote about how trials come to India (and other lower/middle income countries) in an exploitative way: “AstraZeneca explains that the reasons for conducting placebo controlled trials with schizophrenic patients exclusively in low- and middle incomes countries is because “almost all Western ethics committees do not approve this kind of trials anymore because of ethical concerns and therefore AstraZeneca is compelled to look for destinations outside Western Europe as these placebo controlled studies are still required by the EMEA and the FDA for market authorisation” (‘Putting Contract Research Organisations on the Radar’, Mariëtte van Huijstee & Irene Schipper, 2011). 

2. “We provide an example of ethics dumping in three trials conducted from 1998 to 2015 in urban and rural India on testing for cervical cancer... These trials have been condemned as unethical by public health experts and ethicists as the participants were not offered the same level of protection and consideration as participants in high income countries would have been.” (Schroeder D, Cook Lucas J, Fenet S, Hirsch F (eds) (2016) “Ethics Dumping” – Paradigmatic Case Studies, a report for TRUST)

- - - - - - - - - - - - - - - - - - - - - - - - - - - - - - - - - - - - - - - - - - 

3. There are definitely problems with the CTRI website and its updation. Are these problems only for multinational trials OR are also seen in Indian trials. This is important before we infer malafide intentions.

[We completely agree that CTRI’s problems relate to all trials, not just those sponsored by multinationals. We are not inferring malafide intentions of the CTRI staff, or CDSCO. In 2019, we published a 17-page paper (Pillamarapu et al. Trials.) on some of the major categories of problems with CTRI and its data. Similar problems are found in public registries in other countries as well, as documented in that paper. Highlighting problems can lead to remedial action, and therefore we believe there is value in it. We quote (for the reviewer, not in the manuscript) from a PhD thesis from Oxford University, on the issue of clinical trial registries: “In the absence of statutory enforcement, open public audit is widely recognised as a valuable tool to increase accountability and improve quality.433,434] (DeVito, N., 2022, Trial Registries for Transparency and Accountability in Clinical Research. PhD thesis. Available at https://ora.ox.ac.uk/objects/uuid:3c901316-5c24-4b42-a78e-1037d56d35e4)

We have added the following line to the Conclusion: “Nor are we inferring malafide intentions of the CTRI staff, or of CDSCO.”

- - - - - - - - - - - - - - - - - - - - - - - - - - - - - - - - - - - - - - - - - - 

Comments:

4. Title: needs rephrasing as “some” should not be cause of concern. Better to be neutral – “Representation from India in multinational trials registered in national trial registry”.

[We have changed the title to ‘Representation from India in multinational, interventional, Phase 2 or 3 trials registered in Clinical Trials Registry-India: A cross-sectional study’]

- - - - - - - - - - - - - - - - - - - - - - - - - - - - - - - - - - - - - - - - - - 

5. Authors do not explain how they defined unfair burden-benefit ratio. Similarly, they need to define their main outcome variable (participation from India) and its measurement in the methods section.

[In the Discussion, we have now explained our belief that ‘60% participation from India’ is a limit to the fraction that is acceptable in terms of burden-benefit ratio. And in the Methods section we have defined the main outcome variable.]

- - - - - - - - - - - - - - - - - - - - - - - - - - - - - - - - - - - - - - - - - - 

6. There is poor differentiation between methods/ results and discussion. Needs to be rewritten. Results starts with a statement – we posed three questions which is clearly a methods component.

[We were not sure how to proceed with this comment. If (as stated below in Reviewer Comment 8), Figure 1 is a part of Results and not Methods, then it is difficult to move the ‘we posed three questions’ para to the Methods since these questions arose from the finding of the two sets of data (earlier named Set A and Set B and now named the ‘Completed Set’, and the ‘Not Completed Set’, respectively, in response to Reviewer 2’s request to make the naming more informative). For that reason we have not moved this para from Results to Methods. However we have moved Figure 1.]

- - - - - - - - - - - - - - - - - - - - - - - - - - - - - - - - - - - - - - - - - -

7. Issue of Sanofi Pasteur and Shanta Biotech – if at all it needs to be in the paper, it should be in discussion.

[We have moved these details to the S8 File where more details are provided about each trial.]

- - - - - - - - - - - - - - - - - - - - - - - - - - - - - - - - - - - - - - - - - - 

8. For example, figure 1 is a part of result and not methods.

[We have moved Figure 1 to the results.]

- - - - - - - - - - - - - - - - - - - - - - - - - - - - - - - - - - - - - - - - - - 

9. Might be better to give details of the trials (table 1) in supplement.

[Table 1 details are already available in S7 File. We have deleted Table 1.]

- - - - - - - - - - - - - - - - - - - - - - - - - - - - - - - - - - - - - - - - - - 

10. There are many arbitrary definitions and cut-offs which need to be justified.

[In the Discussion, we have now addressed the issue of (a) 60% as an acceptable cut off for recruitment from India; (b) the categories of what could or could not be considered an acceptable increase in the fraction of recruitment from India, and (c) for trials without a ‘Date of Study Completion’, the various assumptions made in order to calculate whether or not a trial is likely to have been completed.]

- - - - - - - - - - - - - - - - - - - - - - - - - - - - - - - - - - - - - - - - - - 

11. 60% is an example. 60% or more actual recruitment – was this 60% out of total or 60% more than the planned (denominator is the planned number).

[It is 60% of the total. We have now clarified what the denominator is for various percentages. throughout the manuscript.]

- - - - - - - - - - - - - - - - - - - - - - - - - - - - - - - - - - - - - - - - - - 

12. I would not put not studying before 2013 as a limitation.

[We have removed this limitation.]

- - - - - - - - - - - - - - - - - - - - - - - - - - - - - - - - - - - - - - - - - - 

13. Figure 1. Should be like a typical flow charts and explain exclusions as well. It would be useful if data on all phases of trials are provided before picking phase 2-3 trials.

[Figure 1 has been redone, with the numbers for, and descriptions of, all the trials that were excluded.]

- - - - - - - - - - - - - - - - - - - - - - - - - - - - - - - - - - - - - - - - - - 

Reviewer #2: 

14. The paper needs a structured abstract, and the raw numbers in the abstract would benefit from percentages.

[a. Regarding the structured abstract, in the ‘Instructions to authors’, the relevant points about the Abstract are:

 • Describe the main objective(s) of the study 

 • Explain how the study was done, including any model organisms used, without methodological detail 

 • Summarize the most important results and their significance.

There is no mention of sub-headings in an Abstract, and when we checked a few recent papers on the PLOS ONE site, most were in the format of a single paragraph. We have been in a dilemma as to what to do, and have kept the current format.]

[b. We have included percentages both in the Abstract and in the main text.]

- - - - - - - - - - - - - - - - - - - - - - - - - - - - - - - - - - - - - - - - - - 

15. The title might need to give a clearer sense of the types of trials included and should probably give a sense of the study design for the interested reader.

[We have changed the title to ‘Representation from India in multinational, interventional, Phase 2 or 3 trials registered in Clinical Trials Registry-India: A cross-sectional study’]

- - - - - - - - - - - - - - - - - - - - - - - - - - - - - - - - - - - - - - - - - - 

16. I would move the reason for the eight year period to the methods, as I felt that justification for that was only introduced late in the paper.

[We have done this.]

- - - - - - - - - - - - - - - - - - - - - - - - - - - - - - - - - - - - - - - - - - 

17. What was an ‘ambiguous’ trial (page 5). Some examples might be useful here as there are lots of these and they are not really considered further in the analysis.

[We have now provided 3 examples of the types of ambiguities in the Results section.

All 384 cases are detailed in S6 File.xls. Based on this comment by Reviewer 2, we revisited S6 File, and subdivided the 384 cases into multiple sheets based on the type of ambiguity, as follows.

Category of problem

Number of trials

 1. 

Entirely foreign trials.

22

 2. 

Terminated in one or both fields, ‘Recruitment Status (Global)’ or ‘Recruitment Status (India)’.

77

 3. 

‘Recruitment Status (India)’ was ‘Suspended’.

1

 4. 

‘Total Sample Size’ and ‘Sample Size from India’ are the same, and therefore they appear to be Indian trials. However information in either ‘Recruitment Status_Global’ or ‘Date of first enrollment_Global’ is inconsistent with an Indian trial.

256

 5. 

‘Total Sample Size’ exceeds ‘Sample Size from India’ and therefore they appear to be multinational trials. However for ‘Recruitment Status_Global’, all the trials say ‘Not applicable’.

18

 6. 

Other trials

10

Based on this data, we have reworded the sentences as follows.

Original sentences: Based on these criteria, we obtained 5580 Indian studies, 473 multinational ones and 384 ambiguous cases (S6 File). 

New sentences: BBased on these criteria, we obtained 5580 Indian studies, 473 multinational ones and 384 other trials (S6 File). Of the 384 other cases, 22 were entirely foreign trials; 77 were terminated either in India or elsewhere; 1 was suspended in India; 256 appeared to be Indian trials based on data in some fields, but had ambiguous information in other fields; 18 appeared to be multinational trials based on data in some fields, but had ambiguous information in other fields; and 10 trials had other kinds of discrepancies. We provide three examples from these 384 cases in Table 1.]

- - - - - - - - - - - - - - - - - - - - - - - - - - - - - - - - - - - - - - - - - - 

The study is largely desccriptive and I think the methods are sensible and appropriate, with the main issues around the exact procedures used to assign trials to various categories.

- - - - - - - - - - - - - - - - - - - - - - - - - - - - - - - - - - - - - - - - - - 

18. I do think the reason for the 60% criterion needs justification. As they highlight, it may be too low, but they need to give a sense of the reasons behind that threshold, as it is not clear why that was chosen.

We have added the following para to the Discussion:

[Trials come with risks for the participants. They also come with potential (a) direct benefit to the participants, or (b) indirect benefit to those of the same ethnicity in case the intervention is subsequently approved and is affordable and accessible to similar populations. In a multinational trial, where there is at least one country other than India, there ought to be a fair distribution of both the risks and potential benefits to the participants in the participating countries. As such, the fraction of participants from India should never be more than 50%, and proportionately lower based on how many countries participate in the trial. However, given that India has almost 1.4 billion people, it is almost a certainty that there will be more patients with a given condition or disease in India than in any other country, except perhaps China. Logistically speaking, it ought to be easier to conduct a trial in fewer countries, provided there are enough patients. So, in the interest of each trial being conducted in a realistic time-frame, and without the extra costs of setting up sites in too many countries, we believe that recruiting 60% of the participants from India is ethical. ]

- - - - - - - - - - - - - - - - - - - - - - - - - - - - - - - - - - - - - - - - - - 

19. There is a good argument for some sensitivity analyses around this figure, since it is so important. To avoid confusing the main message, they might be included as an ‘additional file’ for interested readers to show how different thresholds impact on the results.

[This has been done, and suitable information added in the main manuscript.]

- - - - - - - - - - - - - - - - - - - - - - - - - - - - - - - - - - - - - - - - - - 

20. Labelling (and naming) studies as ‘problematic’ or ‘ethically questionable’ may be an issue. I assume all these studies have received ethical clearance of some form, and although I do not feel that means that they cannot be reconsidered in terms of ethical issues by other observers, it is quite a leap to apply such labels, without the authors/companies having right of reply. Over-recruitment in particular settings does raise ethical issues, but I do not think it is by definition ‘unethical’ or ‘questionable’ or ‘unacceptable’ (which are quite loaded terms), and I think that is an important distinction. I would use less emotive language in labelling. The ethical issues can be highlighted in the introduction and discussion.

[We deleted Table 1 in response to Reviewer Comment 9. We have removed the ‘Comment’ section from Table 3. Ethical issues are now mentioned only in the Introduction and Discussion.]

- - - - - - - - - - - - - - - - - - - - - - - - - - - - - - - - - - - - - - - - - - 

21. The labelling of studies as Set A and Set B is not very helpful, and it would be much better to have succinct but meaningful labels for these based on the actual criteria used to define them.

[Set A is now the ‘Completed Set’, and Set B is the ‘Not Completed Set’]

- - - - - - - - - - - - - - - - - - - - - - - - - - - - - - - - - - - - - - - - - -

---

## [Decision Letter · Decision Letter 1]

18 Oct 2022

PONE-D-22-19988R1

Representation from India in multinational, interventional, Phase 2 or 3 trials registered in Clinical Trials Registry-India: A cross-sectional study

PLOS ONE

Dear Dr. Saberwal,

Thank you for submitting your manuscript to PLOS ONE. After careful consideration, we have decided that your manuscript does not meet our criteria for publication and must therefore be rejected.

I am sorry that we cannot be more positive on this occasion, but hope that you appreciate the reasons for this decision.

Kind regards,

Tarik A. Rashid, PhD

Academic Editor

PLOS ONE

Additional Editor Comments:

The paper is rejected by reviewer 1 based on the following points of view:

1-While I share the concerns of the authors, I would prefer to adopt a non-judgemental approach to writing the paper and focus on providing objective view. They could voice their concerns in the discussion but not throughout the paper.

2-My second concern is that the authors say that their prime concern is the Indian representation, while the major section of the paper is on information available in CTRI. Either they broaden the focus on information on International trials in CTRI portal and define clear indicators for its evaluation. Right now it is a hotch-potch. Finally the data on India representation comes from very few trials.

3- Authors need to strike a balance as not having enough Indians will also be criticized in todays world and having too much as well. Also I do not agree, that trials should be only for public health important diseases as this would mean rare disease are not studied and they may also be higher numbers in INdia.

4-There are four major stakeholders in this issues - trial sponsors, IEC/IRBs, Drug regulatory authorities and ICMR-CTRI. Authors need to delineate clearly where each of the them is not doing their job.

5- I would also expect them to avoid arbitrary cut-offs and just present the distribution wherever possible. Key indicators could be planned enrollment, actual enrollment (%) and ratio of them.

Reviewers' comments:

Reviewer's Responses to Questions

**Comments to the Author**

1. If the authors have adequately addressed your comments raised in a previous round of review and you feel that this manuscript is now acceptable for publication, you may indicate that here to bypass the “Comments to the Author” section, enter your conflict of interest statement in the “Confidential to Editor” section, and submit your "Accept" recommendation.

Reviewer #1: (No Response)

Reviewer #2: (No Response)

2. Is the manuscript technically sound, and do the data support the conclusions?

Reviewer #1: No

Reviewer #2: Partly

3. Has the statistical analysis been performed appropriately and rigorously? 

Reviewer #1: N/A

Reviewer #2: Yes

4. Have the authors made all data underlying the findings in their manuscript fully available?

Reviewer #1: Yes

Reviewer #2: Yes

5. Is the manuscript presented in an intelligible fashion and written in standard English?

Reviewer #1: No

Reviewer #2: Yes

6. Review Comments to the Author

Reviewer #1: While I share the concerns of the authors, I would prefer to adopt a non-judgemental approach to writing the paper and focus on providing objective view. They could voice their concerns in the discussion but not throughout the paper.

My second concern is that the authors say that their prime concern is the Indian representation, while the major section of the paper is on information available in CTRI. Either they broaden the focus on information on International trials in CTRI portal and define clear indicators for its evaluation. Right now it is a hotch-potch. Finally the data on India representation comes from very few trials.

Authors need to strike a balance as not having enough Indians will also be criticized in todays world and having too much as well. Also I do not agree, that trials should be only for public health important diseases as this would mean rare disease are not studied and they may also be higher numbers in INdia.

There are four major stakeholders in this issues - trial sponsors, IEC/IRBs, Drug regulatory authorities and ICMR-CTRI. Authors need to delineate clearly where each of the them is not doing their job.

I would also expect them to avoid arbitrary cut-offs and just present the distribution wherever possible. Key indicators could be planned enrollment, actual enrollment (%) and ratio of them.

Table 1 can be deleted.

Reviewer #2: Thank you for the revised manuscript, which deals with the majority of the issues I raised

However, I would like two minor changes

The justification of the 60% criterion is helpful, but I think that needs to go in the Methods. They can consider the wider issues relating to that threshold and potential criticisms in the Discussion, but I feel that this is a methodological issue and needs to come quite early in the paper, even at the cost of some mild repetition. The abstract also needs to include a statement that defines the 60% threshold

There are also still a few 'loaded' phrases which I do not think are helpful. I am happy that the label their phenomenon as 'potentially problematic' (I do think 'potentially' is required). The comment about 'unfair burden' needs rephrasing because the issue is about the burden on the population, not individual recruits. The statement about 'ethically questionable' should be removed, because all they have done is identified something worthy of further consideration, and are not in a position to make a strong ethical judgement

I also think the reference to 'guinea pigs' on page 4 should be rephrased.

I do not need to see the manuscript again

7. PLOS authors have the option to publish the peer review history of their article (what does this mean?). If published, this will include your full peer review and any attached files.

Reviewer #1: **Yes: **Dr. Anand Krishnan

Reviewer #2: **Yes: **Peter Bower

- - - - -

---

## [Author Response · Author response to Decision Letter 1]

11 Dec 2022

Point-by-point response to the Editor’s and the reviewers’ comments.

- - - - - -- - - - - - - - - - - - - - -

Editor's comments:

The Editor’s comments were a re-iteration of some of Reviewer #1’s comments. As such, all of them have been responded to below.

- - - - - -- - - - - - - - - - - - - - -

Reviewers' comments:

Reviewer #1: 

1. While I share the concerns of the authors, I would prefer to adopt a non-judgemental approach to writing the paper and focus on providing objective view. They could voice their concerns in the discussion but not throughout the paper.

[In the first round of revisions, Reviewer 2 had commented “The ethical issues can be highlighted in the introduction and discussion.”, which we did. We do need to mention our concerns in the Introduction, since they form the rationale for the study. As such, this comment by Reviewer 1 has already been addressed.]

- - - - - -- - - - - - - - - - - - - - -

2. My second concern is that the authors say that their prime concern is the Indian representation, while the major section of the paper is on information available in CTRI. Either they broaden the focus on information on International trials in CTRI portal and define clear indicators for its evaluation. Right now it is a hotch-potch. 

[We have analyzed Indian representation, from data available in CTRI. These records have data fields for both Indian participation and ‘global participation’. For certain analyses, we have used the fields linked to Indian participation and for some global participation. There is nothing hotch-potch about our handling of the data. We are quite used to handling CTRI data, as evidenced by the following publications from our group: Pillamarapu et al, 2019 (doi: 10.1186/S13063-019-3592-0), Kumari et al, 2020 (doi: 10.1371/journal.pone.0234925), Chakraborty and Saberwal, 2022 (doi: 10.20529/IJME.2022.033), Chakraborty et al, 2022 (https://doi.org/10.1371/journal.pgph.0000617).]

- - - - - -- - - - - - - - - - - - - - -

3, Finally the data on India representation comes from very few trials.

[Yes, and that is due to the poor quality of the data in CTRI which has been the focus of some of our published work, such as a 17-page paper, Pillamarapu et al, 2019 An analysis of deficiencies in the data of interventional drug trials registered with Clinical Trials Registry – India. (doi: 10.1186/S13063-019-3592-0). In the current manuscript, we have provided a flowchart that shows why the numbers have been cut down at various stages. Also, in the Conclusion, we have mentioned “Further, the records of several studies that were probably completed were not updated in CTRI in a timely manner.”, which is one of the reasons for the low numbers. We are happy to enhance the Discussion by talking about the poor quality of the records, that leads to these low numbers. We could also add the low numbers as a Limitation.

The data is a quantitative description of a potential problem. Although observers have lamented that participants from India are over-represented (Eg. Pandey et al. Strengthening ethics in clinical research. Indian J Med Res 133, March 2011, pp 339-340. https://www.academia.edu/60714767/Strengthening_ethics_in_clinical_research), to the best of our knowledge, this is the only study that tries to quantify the problem.]

- - - - - -- - - - - - - - - - - - - - -

4. Authors need to strike a balance as not having enough Indians will also be criticized in todays world and having too much as well. 

[We agree that not having enough Indians in multinational trials is a problem, and in fact we have highlighted this problem – for orphan drugs – in a recent publication (Chakraborty et al. Rare disease patients in India are rarely involved in international orphan drug trials. PLOS Global Public Health, doi:10.1371/journal.pgph.0000890, 2022). However the reasons that under-representation and over-representation are problematic are different. In the former, it is that without adequate ethnic diversity, the results may not be generalizable. In the latter, it is the worry of exploitation. In this work, we have focussed on the latter. We would be happy to clarify this point in the Discussion.]

- - - - - -- - - - - - - - - - - - - - -

5. Also I do not agree, that trials should be only for public health important diseases as this would mean rare disease are not studied and they may also be higher numbers in India.

[We have not argued along these lines. All that we are saying is that if a condition or disease is widespread in the world, then there is no ethical justification for over-recruiting from India. If a disease – such as tuberculosis or leishmaniasis – is more prevalent in India, then it is not ethically problematic to recruit a larger fraction of participants from India.]

- - - - - -- - - - - - - - - - - - - - -

6. There are four major stakeholders in this issues - trial sponsors, IEC/IRBs, Drug regulatory authorities and ICMR-CTRI. Authors need to delineate clearly where each of the them is not doing their job.

[Why do we need to do this? It’s as though we are offering an apple, but are being asked to offer an orange instead. Is that fair?

We are merely raising a red flag about possible over-recruitment from India. If the phenomenon is confirmed as a problem, then future research can look into which stakeholder is responsible. Fixing responsibility would be a major exercise, and a non-trivial one, since there is very little data in the public domain that would help with that.]

7. I would also expect them to avoid arbitrary cut-offs and just present the distribution wherever possible. Key indicators could be planned enrollment, actual enrollment (%) and ratio of them.

[How does one flag an issue unless one uses cut offs? We have used an arbitrary cut off, have clarified that it is arbitrary, and have also done a sensitivity analysis. We have provided all the data, and readers can come to their own conclusions.]

- - - - - -- - - - - - - - - - - - - - -

8. Table 1 can be deleted.

[Table 1 was created in the first round of revisions, in response to Reviewer 2’s request. We would request the Editor’s advice on whether or not to delete it, since the two reviewers are requesting contradictory actions.]

- - - - - -- - - - - - - - - - - - - - -

Reviewer #2: Thank you for the revised manuscript, which deals with the majority of the issues I raised. However, I would like two minor changes

9. (a) The justification of the 60% criterion is helpful, but I think that needs to go in the Methods. They can consider the wider issues relating to that threshold and potential criticisms in the Discussion, but I feel that this is a methodological issue and needs to come quite early in the paper, even at the cost of some mild repetition. (b) The abstract also needs to include a statement that defines the 60% threshold.

[We have implemented these suggestions. We have also made some edits in the Discussion due to moving some sentences to Methods.]

- - - - - -- - - - - - - - - - - - - - -

10. There are also still a few 'loaded' phrases which I do not think are helpful. I am happy that the label their phenomenon as 'potentially problematic' (I do think 'potentially' is required). The comment about 'unfair burden' needs rephrasing because the issue is about the burden on the population, not individual recruits. 

[We have reworded a sentence each in the Abstract and Discussion, as follows.]

Abstract: 

Four of the five (7% of 62) had a foreign sponsor, and therefore there was an unfair burden-benefit ratio on Indian recruitees. 

Changed to: Four of the five (7% of 62) had a foreign sponsor, and therefore there was an unfair burden-benefit ratio on the Indian population. 

Discussion: 

Given the history of exploitation of trial participants in India, and given that all trials carry an element of risk, we believe that there is an unfair burden-benefit ratio on Indian recruitees if foreign sponsors, that are conducting trials in multiple countries, recruit too many patients from this country.

Changed to: Given the history of exploitation of trial participants in India, and given that all trials carry an element of risk, we believe that there is an unfair burden-benefit ratio on the Indian population if foreign sponsors, that are conducting trials in multiple countries, recruit too many patients from this country.

- - - - - -- - - - - - - - - - - - - - -

11. The statement about 'ethically questionable' should be removed, because all they have done is identified something worthy of further consideration, and are not in a position to make a strong ethical judgement

In the Abstract and Conclusion, we have deleted ‘Each of these is an ethically questionable practice.’

- - - - - -- - - - - - - - - - - - - - -

12. I also think the reference to 'guinea pigs' on page 4 should be rephrased.

Original line: All of this has led to apprehensions that people in developing countries have sometimes been used as guinea pigs for trials sponsored by organizations in the developed world [6–9].

Changed to: All of this has led to apprehensions that people in developing countries have sometimes been exploited in trials sponsored by organizations in the developed world [6–9].

- - - - - -- - - - - - - - - - - - - - -

---

## [Decision Letter · Decision Letter 2]

24 Feb 2023

PONE-D-22-19988R2

Representation from India in multinational, interventional, Phase 2 or 3 trials registered in Clinical Trials Registry-India: A cross-sectional study

PLOS ONE

Dear Dr. Saberwal,

Thank you for submitting your manuscript to PLOS ONE. After careful consideration, we feel that it has merit but does not fully meet PLOS ONE’s publication criteria as it currently stands. Therefore, we invite you to submit a revised version of the manuscript that addresses the points raised during the review process.

ACADEMIC EDITOR:

Make sure to address the concerns of the third reviewer.

We look forward to receiving your revised manuscript.

Kind regards,

Tarik A. Rashid, PhD

Academic Editor

PLOS ONE

Journal Requirements:

Additional Editor Comments (if provided):

*Comments from PLOS Editorial Office: We note that one or more reviewers has recommended that you cite specific previously published works. As always, we recommend that you please review and evaluate the requested works to determine whether they are relevant and should be cited. It is not a requirement to cite these works. We appreciate your attention to this request.*

Reviewers' comments:

Reviewer's Responses to Questions

**Comments to the Author**

1. If the authors have adequately addressed your comments raised in a previous round of review and you feel that this manuscript is now acceptable for publication, you may indicate that here to bypass the “Comments to the Author” section, enter your conflict of interest statement in the “Confidential to Editor” section, and submit your "Accept" recommendation.

Reviewer #2: All comments have been addressed

Reviewer #3: All comments have been addressed

2. Is the manuscript technically sound, and do the data support the conclusions?

Reviewer #2: Yes

Reviewer #3: Yes

3. Has the statistical analysis been performed appropriately and rigorously? 

Reviewer #2: Yes

Reviewer #3: Yes

4. Have the authors made all data underlying the findings in their manuscript fully available?

Reviewer #2: Yes

Reviewer #3: Yes

5. Is the manuscript presented in an intelligible fashion and written in standard English?

Reviewer #2: Yes

Reviewer #3: Yes

6. Review Comments to the Author

Reviewer #2: The authors have taken a somewhat combative attitude to the suggestions of reviewer 1, so it will be interesting to see if they are happy with the changes and the areas where the authors have not made changes requested

However, with respect to the changes I have requested, I am happy with the revisions

Reviewer #3: The study is very important, I suggest publishing it. Some minor comments:

1. Introduction: The objective of the study should be re-written, so it can help the reader to know the novelty of the study.

2. Some references were too old, I suggest the author update these references to the latest years. There were also several references which analyzed the Clinical Trials Registry, I think these references will help the authors. Such as: PMID: 33042806; PMID: 30863312; PMID: 35881593

7. PLOS authors have the option to publish the peer review history of their article (what does this mean?). If published, this will include your full peer review and any attached files.

Reviewer #2: **Yes: **Peter Bower

Reviewer #3: No

---

## [Author Response · Author response to Decision Letter 2]

27 Feb 2023

ACADEMIC EDITOR:

Make sure to address the concerns of the third reviewer.

[Authors: We have done that, as detailed below.]

- - - - - -- - - - - - - - - - - - - - -

Reviewer #3: 

1. Introduction: The objective of the study should be re-written, so it can help the reader to know the novelty of the study.

[Authors: We have done that, both in the Abstract and in the Introduction.]

- - - - - -- - - - - - - - - - - - - - -

2. Some references were too old, I suggest the author update these references to the latest years. There were also several references which analyzed the Clinical Trials Registry, I think these references will help the authors. Such as: PMID: 33042806; PMID: 30863312; PMID: 35881593

[Authors: We have carefully looked into the oldest references that we have cited. We list the 3 that are before 2010 (arbitrary cut off), below. Each of them is a highly cited article, with 227, 395 and 822 citations each. We therefore respectfully request that we may retain the 1st and 3rd reference. Since one of the suggested references is a suitable replacement for the 2nd, we have substituted it.]

 1. Nundy S, Gulhati CM. A New Colonialism? — Conducting Clinical Trials in India. N Engl J Med. 2005;352(16): 1633–1636. doi:10.1056/NEJMp048361 

[This has been cited 227 times.]

 2. Thiers FA, Sinskey AJ, Berndt ER. Trends in the globalization of clinical trials. Nat Rev Drug Discov. 2008;7: 13–14. doi:10.1038/nrd2441 

[This has been cited 395 times.]

 3. Glickman SW, Cairns CB, Schulman KA. Ethical and Scientific Implications of the Globalization of Clinical Research. N Engl J Med. 2009;360(8): 816–23. doi: 10.1056/NEJMsb0803929 

[This has been cited 822 times.]

We have also included two of the three recommended references, as detailed below:

PMID: 33042806: Dong J, Geng Y, Lu D, Li B, Tian L, Lin D, Zhang Y. Clinical Trials for Artificial Intelligence in Cancer Diagnosis: A Cross-Sectional Study of Registered Trials in ClinicalTrials.gov. Front Oncol. 2020 Sep 15;10:1629. doi: 10.3389/fonc.2020.01629. [We could not determine a link between our manuscript and this reference, and therefore have not included it.]

PMID: 30863312: Chen L, Su Y, Quan L, Zhang Y, Du L. Clinical Trials Focusing on Drug Control and Prevention of Ventilator-Associated Pneumonia: A Comprehensive Analysis of Trials Registered on ClinicalTrials.gov. Front Pharmacol. 2019 Feb 26;9:1574. doi: 10.3389/fphar.2018.01574 [We have included this reference.]

PMID: 35881593: Brøgger-Mikkelsen M, Zibert JR, Andersen AD, Lassen U, Hædersdal M, Ali Z, Thomsen SF. Changes in key recruitment performance metrics from 2008-2019 in industry-sponsored phase III clinical trials registered at ClinicalTrials.gov. PLoS One. 2022 Jul 26;17(7):e0271819. doi: 10.1371/journal.pone.0271819 [We have included this reference.]

---

## [Editor Report · Decision Letter 3]

3 Apr 2023

Representation from India in multinational, interventional, Phase 2 or 3 trials registered in Clinical Trials Registry-India: A cross-sectional study

PONE-D-22-19988R3

Dear Dr. Saberwal,

We’re pleased to inform you that your manuscript has been judged scientifically suitable for publication and will be formally accepted for publication once it meets all outstanding technical requirements.

Kind regards,

Tarik A. Rashid, PhD

Academic Editor

PLOS ONE
---

## [Editor Report · Acceptance letter]

10 Sep 2023

PONE-D-22-19988R3 

Representation from India in multinational, interventional, Phase 2 or 3 trials registered in Clinical Trials Registry-India: A cross-sectional study 

Dear Dr. Saberwal:

I'm pleased to inform you that your manuscript has been deemed suitable for publication in PLOS ONE. Congratulations! Your manuscript is now with our production department. 

Kind regards, 

on behalf of

Dr. Tarik A. Rashid 

Academic Editor

PLOS ONE